# Response Times and Tax Compliance

**Ho Fai Chan** [1,2] **, Uwe Dulleck** [1,2,3] **and Benno Torgler** [1,2,4,*]

1   School of Economics and Finance, Queensland University of Technology, Brisbane, QLD 4000, Australia; hofai.chan@qut.edu.au (H.F.C); uwe.dulleck@qut.edu.au (U.D.)
2   Centre for Behavioural Economics, Society and Technology, Brisbane, QLD 4000, Australia
3   Crawford School of Public Policy, Australian National University, Canberra, ACT 4000, Australia
4   Center for Research in Economics, Management and the Arts, CH-8008 Zürich, Switzerland
*   Correspondence: benno.torgler@qut.edu.au

**Abstract:** Inspired by the work of Rubinstein, this study revisits data from a previous lab experiment to explore the relation between response times and tax compliance and understand the potential non-linearity between them by classifying decisions and individuals into compliance types. We find that individuals' decision response time is related to their compliance decisions. Full-non compliant individuals (those who did not declare any earned income) have shorter response times than those who fully or partially complied. Full-compliant individuals also tend to declare income faster than partially compliant subjects. Such results are robust throughout time and when controlling for contextual characteristics of experimental design. We find non-linearity via an inverted *U*-shape function that reaches its maximum declaration time around a compliance rate of 60%, even after controlling for contextual experimental design factors. In addition, we observe a non-linear relation between cognitive skills, response time, and tax compliance. Participants with relatively high cognitive skills and very low or very high tax compliance level have low response times, while subjects with relatively lower cognitive skills tend to report higher decision times for higher compliance levels.

**Keywords:** response time; tax compliance; cognitive skill; income declaration; tax experiment

---

## 1. Introduction

Following the pioneering laboratory work by Wilhelm Wundt, the topic of response or reaction times has occupied psychology research for 150 years: since the beginning of modern psychology. Wundt created an experimental lab in Leipzig that became "the place" for researchers and students from around the world to learn how to do experimental research in psychology [1]. Influenced by the psychophysical and physiological methods of scholars, such as Fenchner, Helmholtz, and Donders (who measured the speed of nerve and mental events), Wundt applied the reaction time method in his own laboratory work to measure the speed of mental processes. Benjamin [1] refers to Sokal's compilation of previously unpublished documents that include of James Cattell[1]'s journal and letters during the time he was working at Wundt's lab in Germany. Cattell's letters to his parents in January 1884 are indicative of the excitement and challenges that are involved with studying reaction times: "We work in a new field, where others will follow us, who must use or correct our results. We are trying to measure the time that it takes to perform the simplest mental act—as for example to distinguish whether a color is blue or red. As this time seems to be not more than one-hundredth of a second, you can imagine this is no easy task" [1] (p. 42, based on [2]). Such early pioneering endeavors

---

1   Cattell is an important figure in the history of psychology, who not only founded leading journals such *as Psychological Review* of *Psychological Bulletin*, but was also editor of *Science.*

in psychology and psychophysics have influenced the field of mental chronometry [3,4], studying the dynamics of cognition and action via behavioral measures, such as reaction time, response accuracy, and speed-accuracy tradeoffs, to understand the nature of sensation, perception, memory, attention, language, reasoning, or problem solving.

Contrary to psychologists' long history of interest in response time, economists and game theorists have only recently paid close attention to it. Rubinstein [5], for example, puzzles that "[f]or reasons beyond my understanding, economists were hostile to the use of response time until recently, when it became a legitimate and popular tool" (p. 862). In general, Rubinstein's [6] article has had strong influence on using response times in behavioral and experimental economics to investigate the procedural elements of decision making. Although monitoring brain activities is one way of understanding procedural aspects during decision making, he criticized the search for correlations between choices made and activities in various brain centers as being problematic: "[T]his is an expensive and speculative type of research. The technical constraints result in small samples and noisy data and the interpretation of the findings is far from indisputable" (p. 1244). He then argued that there are more obvious physical indicators in game-theoretical settings that produce insights into people's reasoning, such as response times, pointing out that "[v]ery few experimental papers in game theory have reported response times" (p. 1244). Response time as an indicator of the nature or process of the choices in a game has the advantages to be simple and cheap (see also [5]). Rubinstein categorized actions as cognitive (reason process involved), instinctive, and reasonless, stating that more cognitive activity will result in longer response times than instinctive responses in his 2007 paper [6]. Therefore, he adapted the dual-process or dual-system models of judgment that have been prominent and widespread in psychology, even before Kahneman [7] introduced the terminology of system I (automatic, fast, instinctive, not deliberate) and system II (slow, cognitive, systematic, thoughtful, conscious, deliberate) to economics. Alternatively, dual-models have also been classified as "hot" (more automatic and emotional-affective) or "cold" (more controlled and cognitive-deliberate) processes [8]. Such dual classifications are not far removed from Freud's [9] three system classification of id, ego, and superego. Id is instinctive and operates unconsciously, seeking pleasure and avoiding pain. Id is checked by ego, the rational part of the mind mediating between the id and the external world, controlling instincts, but not inhibiting them. Freud's superego even allows inclusions of aspects, such as moral compass, norms, indoctrination, or culture, elements that are often missing in the two systems' classification.

Already, in the 2007 paper [4], Rubinstein used various games, also including the ultimatum game[2]. However, he also acknowledged that in the ultimatum game, "distinguishing between different actions is not straightforward. In particular, it is unclear whether the instinctive action in this case is the 50:50 split or the one in which the proposer demands almost the entire sum" [4] (p. 1253), which led him to justify the use of response time to obtain further clues and insights. He observed that the median response time of those who offered less than $50 (out of $100) was 25% higher than those who offered an equal split. He also checked how individuals acting as responders behave if we assume that the proposer gives them $10 out of $100. Interestingly, he found that both the median response times of those who accepted and those who rejected the $10 were identical, which led him to doubt the fMRI experimental work that attributed the acceptance and rejection of low offers to different sides of the brain (acceptance to the cognitive side, but rejection to the emotional part)[3].

Meanwhile, Rubinstein [5,11] has continued to gather data on response times via his didactic website, extending the number of games and therefore allowing for a better understanding of the

---

[2]　The ultimatum game consists of a two-stage game between two players where the second player (responder) decides whether to accept or reject the offer (split of a total sum amount) proposed by the first player. If accepted, each player gets money based on the offer. If the responder rejects, each player receives nothing.

[3]　See also Dulleck et al. [10] for an exploration of the ultimatum game applying a physiological marker on both, the proposer and the responder.

roles of response times via an interpretation of the meaning of choice in single games, predicting the behavior of players in a game, or defining a typology of players[4].

Inspired by Rubinstein's work, we decided to revisit data from a previous tax compliance experiment [13]. Examining the response times of tax compliance decisions is interesting, as people are faced with the dilemma of whether and how much to cheat. Cheating has received relatively less attention in the response time literature as compared with other dilemmas related to the dictator, ultimatum, or public good games. Jiang [14], for example, studied cheating behaviors in a die-rolling task (participants were able to lie about the side chosen to get higher earnings as the casts were self-reported). The experiments used decision times as a way of measuring the subjects' struggle to be honest between different treatments. Interestingly, as far as we know, tax compliance researchers have not explored response times in detail beyond controlling for it their analysis (see, e.g., [15]). This is surprising given that, 30 years ago, John Carroll [16] already stressed the usefulness of applying process-tracing techniques, such as response times, to obtain insights regarding mental mechanisms when investigating taxpayer compliance.

More recently, it has been criticized that the proposition of distinguishing between intuitive and deliberate choices via an examination of response times has led to backward reasoning [17]. Inferring from fast response times that decisions are intuitive risks ignoring "there is a key distinction between the prediction that an automatic process will occur faster than more deliberative computations, and the classification of a choice as intuitive or automatic because it happens more quickly" [17] (p. 2). Labeling fallacies are common in science, and dual processing research is no exception [18]. Thus, in this paper, we are not going to classify what the response time actually means. We are more interested in understanding the potential of non-linearity between response times and tax compliance. For example, Piovesan and Wengström [19] find, in a modified dictator game, that egoistic decision making via choosing the highest payoff for themselves is faster than social behaviour. Public good games[5] have revealed that fast decision-makers are greater or more generous contributors (for a discussion and criticism of those insights, see [20]). This could indicate that total non-compliance or full compliance are the results of clear-cut heuristics that are faster (low-conflict decisions), while partial compliance might be more likely to reveal higher levels or feelings of conflict [13,21]. As Rubinstein [5] has explored the typology of players, and previous tax compliance research has suggested different types of taxpayers [13,22,23], we will explore the response time differences between different types of taxpayers (classified in our case by difference compliance behaviors).

## 2. Decision Process

Details of the experimental design and setting can be found in Dulleck et al. [13]. The Queensland University of Technology Faculty Research Ethics Advisory Board reviewed the experiment, confirming that it met the requirements of the National Statement on Ethical Conduct in Human Research. The experiment was conducted with a design structure that closely followed other laboratory studies on tax compliance. We will only focus on two main aspects of the Dulleck et al. experiment [13], namely the voluntary income tax reporting decision based on the percentage of earned income declared and the response time during the income declaration, to explore response times in this study. As the emphasis here is on the response time, we are not interested in any particular features of the experiment, such as treatments variations or parameters used in the experiment. For example, the experiment included a public good structure, where the total tax paid by individuals in a group was increased by a factor

---

[4]　For a detailed discussion on the benefits, challenges, and possibilities of exploring response time in cognitive psychology and experimental or behavioural economics see Spiliopoulos and Ortmann [12].

[5]　The public goods game models a collective contribution problem in which individuals can decide how much to contribute towards the group. The collective contribution is multiplied by a certain factor and then equally shared among the individuals of the group. In the experiment all the taxes paid by individuals in a group were multiplied with three different factors (0 = no public good, 1, and 2) being equally redistributed to the group members.

before being equally redistributed among the group (see, [24–27]). We will use a "brute force" approach in the regression analysis using group fixed effects to deal with potential variations in response time due to such heterogeneity (i.e., intercept for each of the 45 experimental groups of four participants). We are also not interested in exploring how the response times are linked to any individual experiences throughout the experiment such as being audited or fined[6]. The dynamics of the experiment are controlled for with time fixed effects (i.e., intercept for each of the 16 experimental rounds).

Figure 1 shows income tax reporting. Participants received income, declared their income to an experimental "tax authority", and then paid taxes on the declared income received in each of the 16 rounds. Tax compliance is measured as the proportion of income declared (ratio of declared to actual income). The response time (in seconds) is measured from the moment participants saw the declaration screen until they pressed the okay button. Therefore, we interchangeably use the word response time with declaration time. Participants were not informed that response time was recorded, nor was response time mentioned in the experiment design or instruction. It should be noted that the subjects did not face any time constraints when making the income declaration (endogenously arising response time). This could mean that participants at the extreme level of compliance or non-compliance respond quickly, not because they decide more intuitively, but rather because the decision is an easier one for them to make. Fast responses are also fast mistakes due to not paying attention or lacking motivation [19]. However, Recalde et al. [19] point out that the concern regarding errors is greater in one-shot interactions and, in addition, we will show that we obtain the same results throughout all the rounds in the experiment.

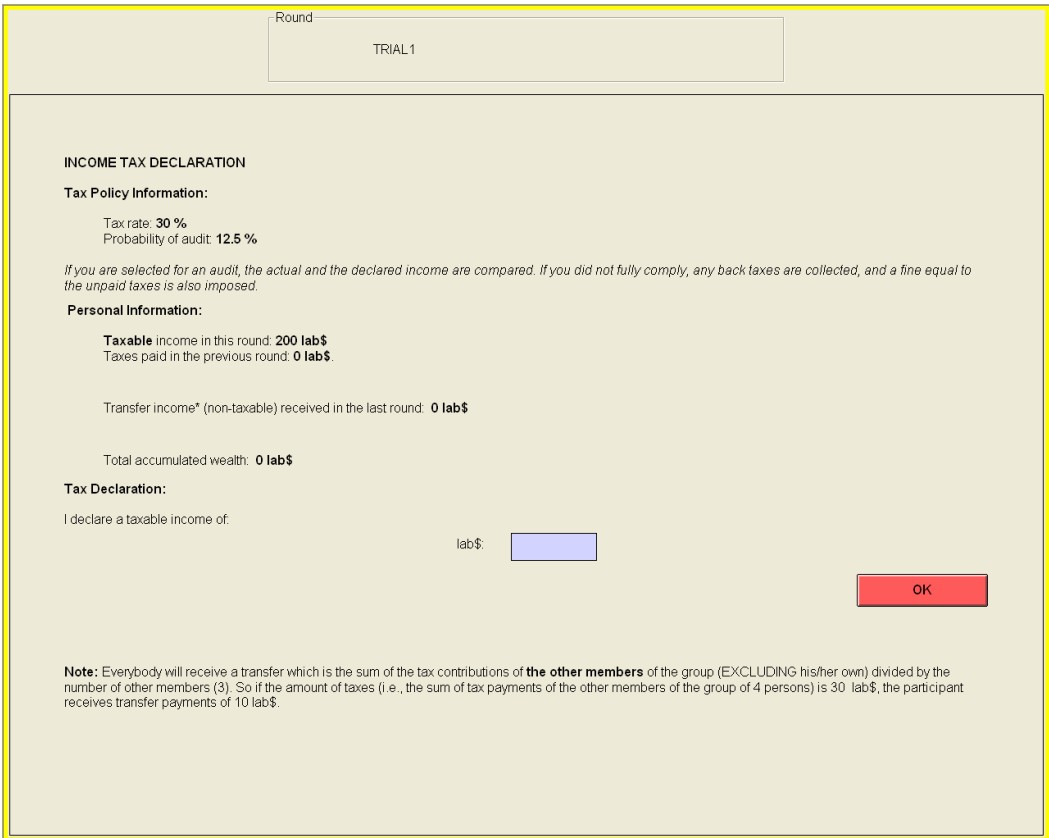

**Figure 1.** Income Tax Declaration (see [13], p. 17).

---

[6]　Nevertheless, we examined the effect of being audited or fined in the previous round and find that it does reduce respond time (on average by ~2 s), after controlling for subject fixed effects).

It should also be noted that we chose to use the data from Dulleck et al. [13] due to the reliable quality of the measurement of the response time. The problem is that computer-based experiments are prone to variation that is caused by network traffic issues that can affect the time recording of events, such as when a screen with the information is visible to the participants. Dulleck et al. [13] isolated the laboratory network from the university and assigned its own subnet to reduce such a problem. The recording discrepancy was reduced to less than six milliseconds.

## 3. Results

In general, we perform two groupings of tax compliance decisions: (1) We look at each individual single declaration decision (2880 decisions, by the amount of earned income declared); and, (2) we look at individual averages out of all the 16 rounds (180 individuals, by the average amount of earned income declared over the entire experiment). We impose six levels of grouping, i.e., full non-compliant (declaring no earned income), full compliant (declared all earned income), and the remaining four partially compliant levels in equally spaced intervals. Table 1 reports the classification of various types of tax compliance and distributions. Not surprisingly, the share of full compliance or no compliance is lower when looking at individual averages, but the results show that there are individuals who are always or never compliant. Table 1 reports that the overall median declaration time is non-linearly related to the level of tax compliance, in both individual-decisions and individual-average grouping, with the highest and lowest compliant levels having the shortest declaration time. We also see that the median declaration time is shorter in the second half of the experiment as compared to the first half, for all tax compliance levels. We report median declaration time, as its distribution is (right) skewed (based on individual-decisions, mean = 14.49, median = 10.19, s.d. = 14.1, min = 1.08, max = 152.36, $N = 2880$).

**Table 1.** Distribution of tax compliance level by decisions and individuals.

| Tax Compliance Level | Individual-Decisions | | | | | Individual-Average | | |
|---|---|---|---|---|---|---|---|---|
| | *N* | Percent | Median (s) | 1st half | 2nd half | *N* | Percent | Median (s) |
| Full non-compliant (0%) | 478 | 16.6 | 6.09 | 7.97 | 5.13 | 11 | 6.11 | 5.4 |
| Very low (0–25%) | 516 | 17.92 | 9.88 | 11.68 | 8.25 | 31 | 17.22 | 11.18 |
| Low (25–50%) | 348 | 12.08 | 12.48 | 14.68 | 10.56 | 36 | 20 | 13.37 |
| High (50–75%) | 312 | 10.83 | 14.88 | 16.46 | 10.45 | 36 | 20 | 15.12 |
| Very high (75–100%) | 296 | 10.28 | 12.38 | 15.62 | 10.10 | 45 | 25 | 12.75 |
| Full compliant (100%) | 930 | 32.29 | 9.95 | 12.16 | 8.16 | 21 | 11.67 | 10.72 |
| Total | 2880 | 100 | | | | 180 | 100 | |

Figure 2 shows the declaration time over the 16 experimental rounds. The declaration time decreases throughout the 16 rounds, which is likely due to the fact that participants were more familiarized with the task. The spike in period 9 can be explained due to the experimental structure. Participants experienced a rule change after round 8 (and before round 9) related to the public good redistribution structure (see [13])[7]. This explains why participants took more time in their decision process in round 9. In general, the results report a consistent and robust picture throughout all panels. Full non-compliance actions or individuals who never declare any income have shorter declaration times when compared with all the other tax compliance levels. In Table 2, we report the results of paired *t*-tests (paired by round) on the average/median decision time (in each round) to assess whether the differences in declaration times between levels of tax compliance are statistically significant. Each *t*-test compares the sample means of individual between two levels of tax compliance (e.g., Full compliant vs.

---

[7] In the first eight rounds, the redistributed amount for each participant in the public good component of the experiment is based on the total contribution all four members to the group. In the remaining rounds, the calculation of redistribution amount only considered the total contribution of the other three members of the group. We also noted a larger impact on decision time for groups all groups except when full non-compliance was observed (see red line Figure 2).

Full non-compliant) over rounds, which means that each test has 32 observations (16 rounds for each comparison pair). The response times are significantly lower for full non-compliance when compared to all other groupings ($p < 0.001$ for all comparisons). Full compliance actions or being always fully compliant are also connected with a tendency for lower response times, but the differences to other compliance groups are not as strong (statistically significant lower response times as compared with most of the partial compliant cases except, for the very low compliance grouping (0–25% of compliance)). Higher levels of partial non-compliance seemed to trigger faster responses than higher levels of partial compliance. Overall, the results are more robust for individual decisions (see panel a and c) due to the larger number of observations.

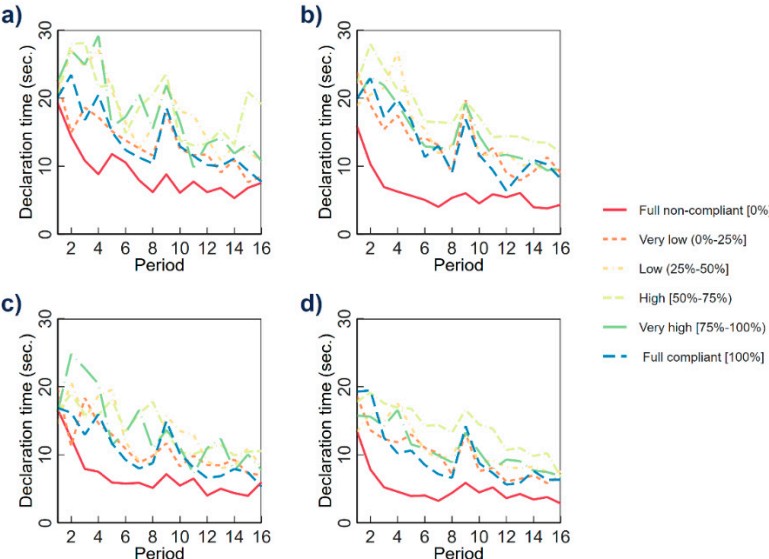

**Figure 2.** Declaration time and tax compliance level over time. (**a**) Average declaration time by tax compliance level of individual decisions; (**b**) Average declaration time by individuals grouped by their average tax compliance level; (**c**) Median declaration time by tax compliance level of individual decisions; and, (**d**) Median declaration time by individuals grouped by their average tax compliance level.

In Figure 3, we further explore the distribution of declaration time by different levels of tax compliance, showing the cumulative percentage frequency. The empirical CDF is shown in base-10 log scale to reduce the skewness. We also differentiate between the first half of the experiment (first 8 rounds, see panel c) and the rounds in the second half (period 9 to 16, see d) due to the spike in round 9. For the declaration time of individual decisions, we use the Two-sample Kolmogorov–Smirnov test to test the equality of distributions between different levels of tax compliance (see Table 3). The significance levels are adjusted for multiple comparisons using the Bonferroni method. Previous results are confirmed, with full non-compliance always returning the fastest declaration times. Full compliance shows shorter response times when compared to partial compliance, except for very low compliance levels (the difference is not statistically significant).

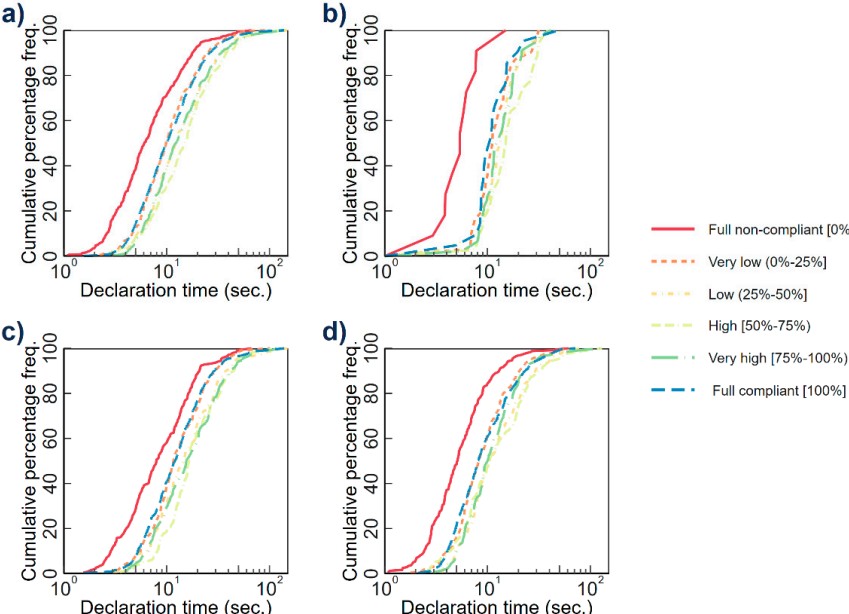

**Figure 3.** Declaration time and tax compliance level. (**a**) Declaration time of individual decisions by the level of tax compliance; (**b**) Individual average declaration time by mean tax compliance level; (**c**) individual-level for the first half of the experiment (round 1 to 8); and, (**d**) Individual-level for the second half of the experiment (round 9 to 16). Horizontal axis (declaration time) is in base-10 log scale.

**Table 2.** Two-sample paired *t*-test of declaration time by tax compliance level, for Figure 2a–d.

| Group 1 | Group 2 | (a) Average | | | | | (b) Average | | | | | (c) Median | | | | | (d) Median | | | | |
|---|---|---|---|---|---|---|---|---|---|---|---|---|---|---|---|---|---|---|---|---|---|
| | | diff. | $t$ | $p_{unadj}$ | $p_{Bon}$ | sig. | diff. | $t$ | $p_{unadj}$ | $p_{Bon}$ | sig. | diff. | $t$ | $p_{unadj}$ | $p_{Bon}$ | sig. | diff. | $t$ | $p_{unadj}$ | $p_{Bon}$ | sig. |
| Full non-compliant | Very low | −4.45 | −6.76 | <0.001 | <0.001 | *** | −7.27 | −9.69 | <0.001 | <0.001 | *** | −4.20 | −6.44 | <0.001 | <0.001 | *** | −4.91 | −8.56 | <0.001 | <0.001 | *** |
| Full non-compliant | Low | −8.65 | −7.55 | <0.001 | <0.001 | *** | −8.72 | −7.68 | <0.001 | <0.001 | *** | −5.95 | −6.31 | <0.001 | <0.001 | *** | −6.02 | −7.69 | <0.001 | <0.001 | *** |
| Full non-compliant | High | −10.08 | −9.2 | <0.001 | <0.001 | *** | −11.57 | −13.35 | <0.001 | <0.001 | *** | −6.86 | −8.69 | <0.001 | <0.001 | *** | −8.95 | −12.48 | <0.001 | <0.001 | *** |
| Full non-compliant | Very high | −8.69 | −7.05 | <0.001 | <0.001 | *** | −8.51 | −9.92 | <0.001 | <0.001 | *** | −6.72 | −6.22 | <0.001 | <0.001 | *** | −5.93 | −9.38 | <0.001 | <0.001 | *** |
| Full non-compliant | Full compliant | −4.75 | −5.87 | <0.001 | <0.001 | *** | −7.06 | −7.42 | <0.001 | <0.001 | *** | −3.57 | −5.87 | <0.001 | <0.001 | *** | −4.76 | −7.00 | <0.001 | <0.001 | *** |
| Very low | Full compliant | −0.30 | −0.46 | 0.6514 | 1 | | 0.21 | 0.35 | 0.7299 | 1 | | 0.63 | 1.01 | 0.3271 | 1 | | 0.15 | 0.30 | 0.7719 | 1 | |
| Low | Full compliant | 3.89 | 7.22 | <0.001 | <0.001 | *** | 1.66 | 2.43 | 0.0282 | 0.4233 | | 2.38 | 4.07 | 0.001 | 0.015 | * | 1.27 | 1.72 | 0.1057 | 1 | |
| High | Full compliant | 5.32 | 5.3 | <0.001 | 0.0013 | ** | 4.51 | 9.52 | <0.001 | <0.001 | *** | 3.29 | 4.38 | <0.001 | 0.0081 | ** | 4.19 | 6.13 | <0.001 | <0.001 | *** |
| Very high | Full compliant | 3.93 | 5.38 | <0.001 | 0.0011 | ** | 1.45 | 2.96 | 0.0098 | 0.1468 | | 3.14 | 3.51 | 0.0032 | 0.0478 | * | 1.18 | 1.85 | 0.0837 | 1 | |
| Very low | Low | −4.20 | −4.8 | <0.001 | 0.0035 | ** | −1.45 | −1.73 | 0.1036 | 1 | | −1.75 | −1.87 | 0.0806 | 1 | | −1.12 | −1.96 | 0.0692 | 1 | |
| Very low | High | −5.63 | −4.93 | <0.001 | 0.0027 | ** | −4.30 | −5.6 | <0.001 | <0.001 | *** | −2.66 | −3.29 | 0.005 | 0.0746 | † | −4.04 | −7.76 | <0.001 | <0.001 | *** |
| Very low | Very high | −4.24 | −4.3 | <0.001 | 0.0095 | ** | −1.24 | −1.85 | 0.0841 | 1 | | −2.51 | −2.38 | 0.0312 | 0.4679 | | −1.03 | −2.13 | 0.05 | 0.7506 | |
| Low | High | −1.43 | −1.32 | 0.2062 | 1 | | −2.84 | −4.08 | <0.001 | 0.0147 | * | −0.91 | −1.04 | 0.3166 | 1 | | −2.92 | −6.66 | <0.001 | <0.001 | *** |
| Low | Very high | −0.04 | −0.05 | 0.9641 | 1 | | 0.22 | 0.36 | 0.7219 | 1 | | −0.76 | −0.73 | 0.4778 | 1 | | 0.09 | 0.29 | 0.7787 | 1 | |
| High | Very high | 1.39 | 1.32 | 0.2059 | 1 | | 3.06 | 10.94 | <0.001 | <0.001 | *** | 0.15 | 0.16 | 0.8779 | 1 | | 3.01 | 7.16 | <0.001 | <0.001 | *** |

$N = 32$ in each paired *t*-test (17 rounds). diff. = $mean_1 − mean_2$. $t$ = *t*-test statistic; $p_{unadj}$ = unadjusted *p*-value (two-sided); $p_{Bon}$ = *p*-value with Bonferroni correction for multiple comparison. sig. = significance. † $p < 0.10$; * $p < 0.05$; ** $p < 0.01$; *** $p < 0.001$ based on Bonferroni adjusted *p*-value.

**Table 3.** Two-sample Kolmogorov-Smirnov test of declaration time by tax compliance level, for Figure 3a–d.

| Group 1 | Group 2 | (a) Individual-Decisions | | | | (b) Individuals | | | | (c) Decisions, First Half | | | | (d) Decisions, Second Half | | | |
|---|---|---|---|---|---|---|---|---|---|---|---|---|---|---|---|---|---|
| | | D | $p_{unadj}$ | $p_{Bon}$ | sig. | D | $p_{unadj}$ | $p_{Bon}$ | sig. | D | $p_{unadj}$ | $p_{Bon}$ | sig. | D | $p_{unadj}$ | $p_{Bon}$ | sig. |
| Full non-compliant | Very low | 0.3068 | <0.001 | <0.001 | *** | 0.2724 | <0.001 | <0.001 | *** | 0.3639 | <0.001 | <0.001 | *** | 0.7478 | <0.001 | 0.0034 | ** |
| Full non-compliant | Low | 0.3969 | <0.001 | <0.001 | *** | 0.3236 | <0.001 | <0.001 | *** | 0.4749 | <0.001 | <0.001 | *** | 0.8258 | <0.001 | <0.001 | *** |
| Full non-compliant | High | 0.4254 | <0.001 | <0.001 | *** | 0.417 | <0.001 | <0.001 | *** | 0.418 | <0.001 | <0.001 | *** | 0.8813 | <0.001 | <0.001 | *** |
| Full non-compliant | Very high | 0.3802 | <0.001 | <0.001 | *** | 0.3235 | <0.001 | <0.001 | *** | 0.4462 | <0.001 | <0.001 | *** | 0.8646 | <0.001 | <0.001 | *** |
| Full non-compliant | Full compliant | 0.2666 | <0.001 | <0.001 | *** | 0.2382 | <0.001 | <0.001 | *** | 0.2969 | <0.001 | <0.001 | *** | 0.8139 | <0.001 | 0.0021 | ** |
| Very low | Full compliant | 0.0521 | 0.3291 | 1 | | 0.0597 | 0.6365 | 1 | | 0.0774 | 0.2449 | 1 | | 0.1705 | 0.8599 | 1 | |
| Low | Full compliant | 0.1455 | <0.001 | <0.001 | *** | 0.1508 | 0.0063 | 0.0951 | † | 0.1852 | <0.001 | 0.0051 | ** | 0.3135 | 0.1475 | 1 | |
| High | Full compliant | 0.211 | <0.001 | <0.001 | *** | 0.261 | <0.001 | <0.001 | *** | 0.1778 | 0.0026 | 0.0389 | * | 0.381 | 0.0426 | 0.6386 | |
| Very high | Full compliant | 0.1367 | <0.001 | 0.0068 | ** | 0.1886 | <0.001 | 0.0089 | ** | 0.1797 | 0.0016 | 0.0242 | * | 0.2635 | 0.2732 | 1 | |
| Very low | Low | 0.151 | <0.001 | 0.0023 | ** | 0.1326 | 0.0628 | 0.9419 | | 0.1866 | 0.001 | 0.0152 | * | 0.2195 | 0.3983 | 1 | |
| Very low | High | 0.247 | <0.001 | <0.001 | *** | 0.2776 | <0.001 | <0.001 | *** | 0.1997 | 0.0012 | 0.0185 | * | 0.3118 | 0.0784 | 1 | |
| Very low | Very high | 0.1562 | <0.001 | 0.0031 | ** | 0.1797 | 0.0057 | 0.0852 | † | 0.1683 | 0.0084 | 0.1255 | | 0.1728 | 0.6438 | 1 | |
| Low | High | 0.1162 | 0.0236 | 0.3541 | | 0.1784 | 0.008 | 0.12 | | 0.0955 | 0.4832 | 1 | | 0.1944 | 0.5041 | 1 | |
| Low | Very high | 0.0281 | 0.9998 | 1 | | 0.0836 | 0.6293 | 1 | | 0.096 | 0.4543 | 1 | | 0.1278 | 0.8997 | 1 | |
| High | Very high | 0.1209 | 0.0235 | 0.353 | | 0.1505 | 0.0516 | 0.7737 | | 0.1508 | 0.0802 | 1 | | 0.2056 | 0.3667 | 1 | |

Test of distribution equality (K–S test) is performed on logged values of decision time (based 10). D = Kolmogorov–Smirnov statistic; $p_{unadj}$ = unadjusted $p$-value (two-sided); $p_{Bon}$ = $p$-value with Bonferroni correction for multiple comparison. sig. = significance. † $p < 0.10$; * $p < 0.05$; ** $p < 0.01$; *** $p < 0.001$ based on Bonferroni adjusted $p$-value.

We next conduct Ordinary Linear Regression (OLS) regressions to show that declaration time is non-linearly related to tax compliance (inverted *U*-shape). When using individual declaration time as a dependent variable, we control for group fixed effects to take any design choices, such as treatment variations, within-experiment, and time fixed effects, into account to control for dynamic changes over the process of the experiment (Figure 4a). We also assess within-individual differences by controlling for subject fixed effects to remove individual heterogeneity (Figure 4c). To capture non-linearity, we include tax compliance and its squared term as predictors in the regression models. Figure 4 reports the predictions of the models, which nicely demonstrate the non-linear relationship between response time and level of tax compliance. The declaration time increases as the level of tax compliance increases until around 60% of tax compliance level (turning points of the predicted model are 0.589, 0.611, and 0.659 for panel a, b, and c, respectively), where declaration time starts to decline.

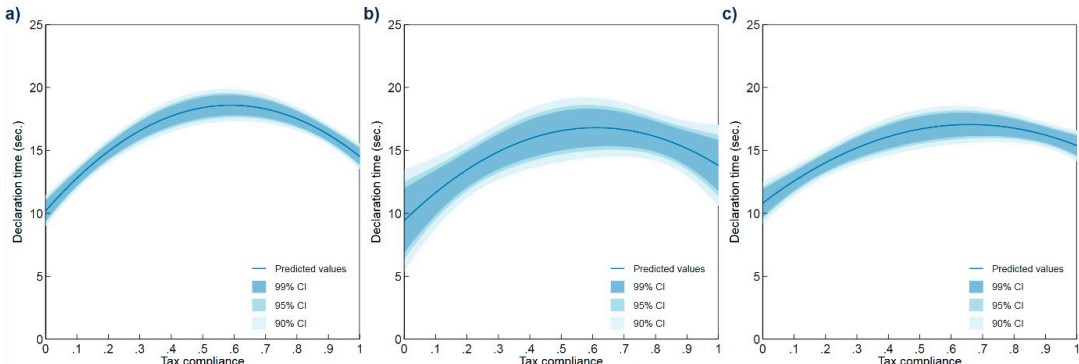

**Figure 4.** Nonlinearity between Declaration Time and Tax Compliance. (**a**) Prediction obtained from Ordinary Linear Regression (OLS) while using declaration time as dependent variable with tax compliance and tax compliance squared as predictors, controlling for experiment group and period fixed effects; (**b**) Prediction obtained from OLS using individual-average declaration time as dependent variable with mean tax compliance and mean tax compliance squared as predictors, controlling for experiment group fixed effects; and, (**c**) Prediction obtained from OLS using declaration time as dependent variable with tax compliance and tax compliance squared as predictors, controlling for subject and period fixed effects.

The Dulleck et al. [13] experiment also included a cognitive skills task comprising 50 questions to be solved within 12 min related to numerical, verbal, and spatial reasoning (see examples in Table A1). We take advantage of the existence of this data to check whether there is a connection between cognitive skill, response time, and tax compliance. The contour plot analysis (Figure 5) is based on the predicted response time that was obtained from the OLS regression model with full interaction between cognitive skill and tax compliance (and its squared term). It shows that participants with higher cognitive skill, in general, have shorter decision times when compared to those with lower cognitive skills. In addition, participants with relatively high cognitive skill with very low or very high tax compliance level exhibit low response times. Decision time increases with higher levels of tax compliance for subjects with lower cognitive skill, although also here we observe some non-linearity.

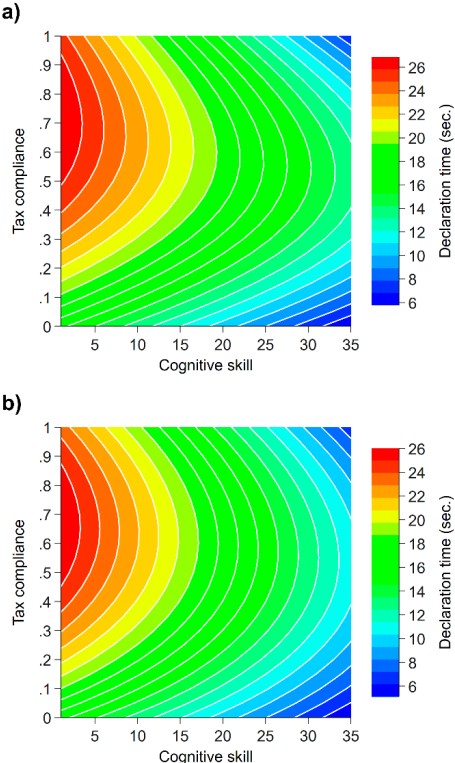

**Figure 5.** Predicted declaration time as a function of tax compliance and cognitive skill. (**a**) Predicted values obtained from Ordinary Linear Regression (OLS) with individual-declaration time as dependent variable and full interaction between tax compliance level of individual-decisions, tax compliance level squared, and cognitive skill as predictors ($N = 2880$); and, (**b**) Predicted values obtained from OLS using individual-average decision time as dependent variable and full interaction between individual-average tax compliance level, tax compliance level squared, and cognitive skill as predictors ($N = 180$).

## 4. Discussion

We find that different levels of tax compliance are connected to different levels of response times based on data from a previous tax compliance experiment [13] well-suited to exploring response times in this context. Extreme behaviors (full non-compliance or full compliance) seemed to facilitate the decision process, possibly due to lower levels of conflicts when making decisions although full evasion is connected with much faster response times than full compliance. For partial tax compliance, the decision time is highest in the range of 60% compliance, which indicates an inverted *U*-shape function in the relationship between response time and tax compliance. Looking at the participants' cognitive skill levels also indicates substantial non-linearity. Participants with relatively high cognitive skill who report very low or very high tax compliance levels have low response times, while those with higher levels of tax compliance but lower cognitive skill show the highest declaration times. In general, we therefore believe that we contribute to the response time literature by taking a much closer look at potential non-linearity. Our results also confirm the usefulness of working with a typology of subjects, although, here, we classify them by degrees of compliance. Nevertheless, results indicate that one could also classify them in line with Rubinstein [5], based on response time information.

## 5. Conclusions

In general, our goal was to show what we can be learned on response times in a tax compliance setting. Future studies could check the robustness of our results. In particular, it would be beneficial to report data from various tax compliance experiments to provide more robust insights as the response time tends to be a noisy variable [11]. Future studies could also test tax declaration decisions that

require more decision time (by reporting, e.g., not only income but also deductions), so as to increase the average response time.

**Author Contributions:** Conceptualization, Writing—Original draft preparation, project administration, B.T.; data curation, visualization, H.F.C.; methodology, formal analysis, investigation, Writing—Review and editing, H.F.C. and B.T.; supervision, U.D. and B.T.

**Funding:** This research received no external funding.

**Acknowledgments:** We thank the Editor and two anonymous referees for helpful remarks and suggestions.

**Conflicts of Interest:** The authors declare no conflict of interest.

## Appendix A

**Table A1.** Cognitive Skill Example Questions.

| Numerical Reasoning | Look at the Row of Numbers. What Number Should Come Next?<br>27　9　3　1　1/3　1/9　1/? |
| --- | --- |
| Verbal reasoning | IMPRISON is the opposite of<br>1. capture, 2. endanger, 3. free, 4. discover, 5. heal |
| | Which figure can be made from the two figures in the brackets? |
| Spatial reasoning |  |

Source: Dulleck et al. [13].

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
