# Peer review of "Response Times and Tax Compliance"

_games, doi:10.3390/g10040045_

Round 1

Reviewer 1 Report

I think this paper is good.  The ethical issue I raise above is minor -- Did the authors get clearance from their Human Subjects Review Board to use the data for this purpose.  I note in the text that students were not told that their response times would be analyzed and as such I suspect that it may be the case that the authors did not request approval to analyse the data in this way.  If this is the case then in my opinion it is up to the Editor as to whether the paper should be published, and if it is whether it should have a disclaimer that this approval was not sought.

The paper is really well written and really well done.  I think the analysis is appropriate.  My only concern is that I am a little surprised that variation in things like being fined in previous periods does not effect reaction time.  But that you can ignore that and still get significant results speaks to the strength of your result.

Author Response

Point 1: I think this paper is good.  The ethical issue I raise above is minor -- Did the authors get clearance from their Human Subjects Review Board to use the data for this purpose.  I note in the text that students were not told that their response times would be analyzed and as such I suspect that it may be the case that the authors did not request approval to analyse the data in this way.  If this is the case then in my opinion it is up to the Editor as to whether the paper should be published, and if it is whether it should have a disclaimer that this approval was not sought.

Response 1: Yes, we refer now that the experiment was reviewed by the ethics advisory board.

Point 2: The paper is really well written and really well done.  I think the analysis is appropriate.  My only concern is that I am a little surprised that variation in things like being fined in previous periods does not effect reaction time.  But that you can ignore that and still get significant results speaks to the strength of your result.

Response 2: We are grateful for the very positive comments from Reviewer 1. The reviewer’s point is interesting in its own right, however, changes in reaction time due to experimental-specific feature (e.g., being audited or receiving a fine) is not within the scope of our study, and as mentioned, we use a “brute force” approach to control these factors using various fixed effects. Nevertheless, given the comment and due to the dynamic nature of the experiment, we examine the effect of being audited or fined in the previous round and find that it does significantly reduce respond time (on average by ~2 seconds), after controlling for subject fixed effects). We report that information now in the paper.

Reviewer 2 Report

Thank you for the opportunity to read your paper “Response Times and Tax Compliance”. The topic is interesting and topical. Results are interesting and logical as well. Nevertheless, I would take the liberty to encourage you to work further on the paper and extend it by, say 20-30% (this is only a rough suggestion). The authors often use an imprecise vague style. Often the meaning of sentences has to be guessed by the reader because of omitted intermediate logic. The very purposes of such an extension are, in my opinion:

too quick shifts to statements or conclusions, which now often appear to be brave mathematical reasoning, again, often appears to be vague, even if there well could have been sufficient grounding sentences appear to be grammatically quite correct, but the text is not read as a sound and understandable one there used different terms instead of adhering to the same terminology through the whole paper. E.g. “response times”, ”response rates”. Both look rather awkward, but the authors may know contemporary terminology. I would most probably use “relation between the taxpayer's response time and the level of tax compliance”, but the authors should decide themselves.

To ground the above suggestions I provide more suggestions hereafter.

The title looks both insufficiently precise and quite awkward. How about the title of the paper could be more detailed as, say, “A ∩-shape relation between taxpayer's response time and the level of tax compliance” or similar? Line 6. Please change “full-non compliant” or explain Line 6 whose “decisions” should be defined (as “their”, for example) Line 6 “are connected” looks vague. How about “related” or “imply” I’d suggest the sentence in lines 5–8 to be re-written. Line 13 the second “with” is redundant Line 27 “from the time he was” [doing what ?] “at Wundt’s lab” Before line 54 at least a short explanation about 2 systems should be inserted Lines 64–75. The experiment was not described sufficiently. Ultimatum game was not described (even if it could be guessed by the reader). Line 63. Article before “backward reasoning” is missing (please, thoroughly re-work the whole text paying also attention to the usage of articles) Lines 101–102. “Public good games” were not defined Line 104. What do you mean by “clear heuristics”? The decision about the level of tax compliance could be heuristic or cognitive in all cases. Or please ground the particular point of view. The argument “One could argue” is not sufficient. Line 112. What do you mean by “public good structure”? Line 114. Simply stating the title of the experiment “voluntary income tax reporting decision” is not enough; it requires at least a short description Lines 114–120. A too vague explanation. Please, be more precise. Lines 121–134. The provided description of the experiment provides insufficient details. 16 rounds should be defined precisely. Is “tax on the declared sum” is a lump-sum tax or income tax? Line 135. “quality” of data was not defined even in broad terms, such as “good” or “reliable” Part 3 should be presented much more precisely. “Non-linearity” of the relationship lacks explanation. Probably, a ∩-shape relation would be more appropriate. 16 periods should be described. Figs a), b), c), d) should be explained in the text. Line 153. Some conclusions from the shape of distribution should be drawn Lines 156–157. It should be described how the “course of experiment” progresses in more detail. It is not clear, why round 9 is particular? Line 159. What do you mean by “public good re-distribution structure” in relation to the topic? Line 161. Do you mean by “Lower declaration times” quicker processing of the tax declaration? Line 165. “Pairs” should be stated precisely. Figure 2. What do you group and why? Please provide a clear description in the text. Table 2. As several lines are identical as “Full non-compliant” and other, you should explain that you take all different pairs. Which level of threshold did you impose prior to the statistical analysis? Table 2 states four levels. Based on which level you draw conclusions? Lines 191–192. Do you test a special kind of distribution or just difference of frequencies? What is the result? Is it rejection or non-rejection of the hypothesis that the distributions are different? How did you reach the conclusion “previous results are confirmed” on line 193? What exactly are you claiming? Please state an explanation in the text. Line 202. Do you strive to reveal just a non-linearity (a vague relationship) or a particular relationship? Lines 203–204. “we control…” is not a clear sentence. Please, state much more precisely, what do you control, and what is the random variable. Variables should be defined both by providing description and notations. Lines 206–207. How do you remove the individual heterogeneity? Lines 207–209. All statements should be much more precise, with explanations. How did you construct Fig. 5? Do you have calculations or interpolation formulae? Table 1 in the Appendix is not relevant to the research

Author Response

Point 1: Thank you for the opportunity to read your paper “Response Times and Tax Compliance”. The topic is interesting and topical. Results are interesting and logical as well.

Response 1: We are grateful for the very positive comments from Reviewer 2.

Point 2: Nevertheless, I would take the liberty to encourage you to work further on the paper and extend it by, say 20-30% (this is only a rough suggestion). The authors often use an imprecise vague style. Often the meaning of sentences has to be guessed by the reader because of omitted intermediate logic. The very purposes of such an extension are, in my opinion: too quick shifts to statements or conclusions, which now often appear to be brave mathematical reasoning, again, often appears to be vague, even if there well could have been sufficient grounding sentences appear to be grammatically quite correct, but the text is not read as a sound and understandable one there used different terms instead of adhering to the same terminology through the whole paper. E.g. “response times”, ”response rates”. Both look rather awkward, but the authors may know contemporary terminology. I would most probably use “relation between the taxpayer's response time and the level of tax compliance”, but the authors should decide themselves.

To ground the above suggestions I provide more suggestions hereafter.

Response 2: We appreciate the comments from the reviewer to increase the clarity of the writing style. We have incorporated most of the suggested changes below. Nevertheless, the guest editor has approved to keep the contribution short and hence we decided to focus on the changes requested without making the contribution substantially longer.

Point 3: The title looks both insufficiently precise and quite awkward. How about the title of the paper could be more detailed as, say, “A ∩-shape relation between taxpayer's response time and the level of tax compliance” or similar?

Response 3: Thank you for the title suggestion. We understand that in other research fields, a more content-driven title is encouraged. However, we decided to adhere to the norm in economics and kept our title short and neat. 

Point 4: Line 6. Please change “full-non compliant” or explain

Response 4: We have added clarification to “full-non compliant” (line 6).

Point 5: Line 6 whose “decisions” should be defined (as “their”, for example)

Response 5: We have incorporated the change.  

Point 6: Line 6 “are connected” looks vague. How about “related” or “imply” I’d suggest the sentence in lines 5–8 to be re-written.

Response 6: We have incorporated the change.  

Point 7: Line 13 the second “with” is redundant.

Response 7: We have incorporated the change.  

Point 8: Line 27 “from the time he was” [doing what ?] “at Wundt’s lab”

Response 8: We have adjusted to the sentence.  

Point 9: Before line 54 at least a short explanation about 2 systems should be inserted

Response 9: A description of the two systems has been provided in Lines 56-58. Additional insights into the two systems can be found in Kahneman’s book that we cited.

Point 10: Lines 64–75. The experiment was not described sufficiently. Ultimatum game was not described (even if it could be guessed by the reader).

Response 10: The concept of ultimatum game should be familiar to readers of Games. Nevertheless, we provide a short description of the ultimatum game in the footnote.

Point 11: Line 63. Article before “backward reasoning” is missing (please, thoroughly re-work the whole text paying also attention to the usage of articles)

Response 11: We believe that the sentence is grammatically correct. We have also thoroughly checked the rest of the text.

Point 12: Lines 101–102. “Public good games” were not defined.

Response 12: The concept of public goods game should be familiar to readers of Games (a standard of experimental economics), but also here we provide a short description of the ultimatum game in the footnote.

Point 13: Line 104. What do you mean by “clear heuristics”? The decision about the level of tax compliance could be heuristic or cognitive in all cases. Or please ground the particular point of view. The argument “One could argue” is not sufficient.

Response 13: We agree with the referee that we need to clarify that sentence which we have done (see lines 104-106).

Point 14: Line 112. What do you mean by “public good structure”?

Response 14: We have added a description of the public good structure used in Dulleck et al. [3].

Point 15: Line 114. Simply stating the title of the experiment “voluntary income tax reporting decision” is not enough; it requires at least a short description

Response 15: We have added a short description on this. A more detailed description of the tax compliance measure can be found in the second paragraph of section 2. We also defer the readers to Dulleck et al. [3] for the full experimental procedure while only highlighting the relevant aspects to the current study.

Point 16: Lines 114–120. A too vague explanation. Please, be more precise.

Response 16: We have adjusted the discussion regarding using fixed effects to control for heterogeneity due to experimental design (treatment variations, parameters, rounds).

Point 17: Lines 121–134. The provided description of the experiment provides insufficient details. 16 rounds should be defined precisely. Is “tax on the declared sum” is a lump-sum tax or income tax?

Response 17: Tax declaration is based on income earned in each round. We have now provided a clearer description.

Point 18: Line 135. “quality” of data was not defined even in broad terms, such as “good” or “reliable”

Response 18: We have incorporated the change.

Point 19: Part 3 should be presented much more precisely. “Non-linearity” of the relationship lacks explanation. Probably, a ∩-shape relation would be more appropriate. 16 periods should be described. Figs a), b), c), d) should be explained in the text.

Response 19: We have added an inverted U-shape relation to the explanation.

Point 20: Line 153. Some conclusions from the shape of distribution should be drawn

Response 20: To deal with the positive skewness of the dependent variable, response time, we report both the mean and median of the measure throughout the analysis. The median is important due to being less affected by skewed (non-symmetric) data than the mean.  

Point 21: Lines 156–157. It should be described how the “course of experiment” progresses in more detail. It is not clear, why round 9 is particular? Line 159. What do you mean by “public good re-distribution structure” in relation to the topic?

Response 21: We have added a footnote describing the change in the public good redistribution structure that happened after round 8.

Point 22: Line 161. Do you mean by “Lower declaration times” quicker processing of the tax declaration?

Response 22: This is correct. We have adjusted to “shorter declaration time”. The definition of declaration time is stated in section 2.

Point 23: Line 165. “Pairs” should be stated precisely.

Response 23: We have incorporated the change accordingly.  

Point 24: Figure 2. What do you group and why? Please provide a clear description in the text.

Response 24: We have now added a description of the tax compliance groupings. The grouping captures full (non-)compliant individuals/decisions, while the rest we divided into 4 groups (equally spaced interval in terms of the level of tax compliance, i.e., 25 percentage points).

Point 25: Table 2. As several lines are identical as “Full non-compliant” and other, you should explain that you take all different pairs. Which level of threshold did you impose prior to the statistical analysis? Table 2 states four levels. Based on which level you draw conclusions?

Response 25: We have clarified the multiple comparisons are between all pair sets. We have also included the level of significance in the discussion.  

Point 26: Lines 191–192. Do you test a special kind of distribution or just difference of frequencies? What is the result? Is it rejection or non-rejection of the hypothesis that the distributions are different? How did you reach the conclusion “previous results are confirmed” on line 193? What exactly are you claiming? Please state an explanation in the text.

Response 26: We use the Two-sample Kolmogorov-Smirnov test to test whether two underlying one-dimensional probability distributions differ. The full results are presented in Table 3. It is conventional that the H0 for the two-sample K-S test hypothesized the two samples come from the same distribution, hence, significance indicates the distributions of the two sample is different.

Point 27: Line 202. Do you strive to reveal just a non-linearity (a vague relationship) or a particular relationship?

Response 27: That is correct, we have now specified that we demonstrate the “inverted U-shape” relationship between decision time and tax compliance level, given the results shown in the earlier section of the paper.

Point 28: Lines 203–204. “we control…” is not a clear sentence. Please, state much more precisely, what do you control, and what is the random variable. Variables should be defined both by providing description and notations.

Response 28: We have now clarified the fixed effects in section 2. Fixed effects are a set of indicator variables (0s and 1s) specific to the level (i.e., for 45 experimental groups and for 16 experimental rounds).

Point 29: Lines 206–207. How do you remove the individual heterogeneity?

Response 29: The subject fixed effects (a set of indicator variables for each participant) absorb any individual-specific unobserved heterogeneity (e.g. effect of gender etc.) that is constant over time (i.e., experiment rounds).

Point 30: Lines 207–209. All statements should be much more precise, with explanations.

Response 30: We have adjusted the statements.

Point 31: How did you construct Fig. 5? Do you have calculations or interpolation formulae?

Response 31: We have now clarified that Fig. 5 (contour plot) is constructed by the predicted values (of response time) from the OLS regression on the full interaction terms between cognitive skills and tax compliance (and its squared term).

Point 32: Table 1 in the Appendix is not relevant to the research.

Response 32: We included Appendix Table 1 for readers’ reference to how cognitive skills were measured.